# ACQUIRED: A Dataset for Answering Counterfactual Questions In Real-Life Videos

**Te-Lin Wu**[*1], **Zi-Yi Dou**[*1], **Qingyuan Hu**[*1], **Yu Hou**[2], **Nischal Chandra**[1],
**Marjorie Freedman**[3], **Ralph Weischedel**[3], **Nanyun Peng**[1],
[1]University of California, Los Angeles, [2]University of Maryland, College Park,
[3]Information Sciences Institute, University of Southern California
{telinwu,zdou,hu528,nrchandra,violetpeng}@cs.ucla.edu
houyu@umd.edu,{mrf,weisched}@isi.edu

## Abstract

Multimodal counterfactual reasoning is a vital ability for AI systems. It involves predicting the outcomes of hypothetical circumstances based on vision and language inputs, which enables AI models to learn from failures and explore hypothetical scenarios. Despite its importance, there are only a few benchmark datasets targeting on evaluating the counterfactual reasoning abilities of multimodal models. Further more, existing datasets either only cover reasoning over synthetic environments, or focus only on specific types of events (e.g. traffic collisions), making them hard to reliably benchmark the model generalization ability in diverse real-world scenarios and reasoning dimensions. To overcome these limitations, we develop a video question answering dataset, ACQUIRED, which consists of 3.7K annotated videos, encompassing a wide range of event types and including both first and third-person viewpoints, ensuring real-world diversity. In addition, each video is annotated with questions that span three distinct dimensions of reasoning, including *physical*, *social*, and *temporal*, which can comprehensively evaluate the model counterfactual abilities along multiple aspects. We benchmark our dataset against several state-of-the-art language-only and multimodal models and experimental results demonstrate a significant performance gap (>13%) between models and humans. The findings suggest that multimodal counterfactual reasoning remains an open challenge and ACQUIRED is a comprehensive and reliable benchmark for inspiring future research in this direction. Our dataset and code are at: https://github.com/PlusLabNLP/acquired

## 1 Introduction

Multimodal counterfactual reasoning refers to the ability to imagine and reason about what might have happened if certain conditions were different

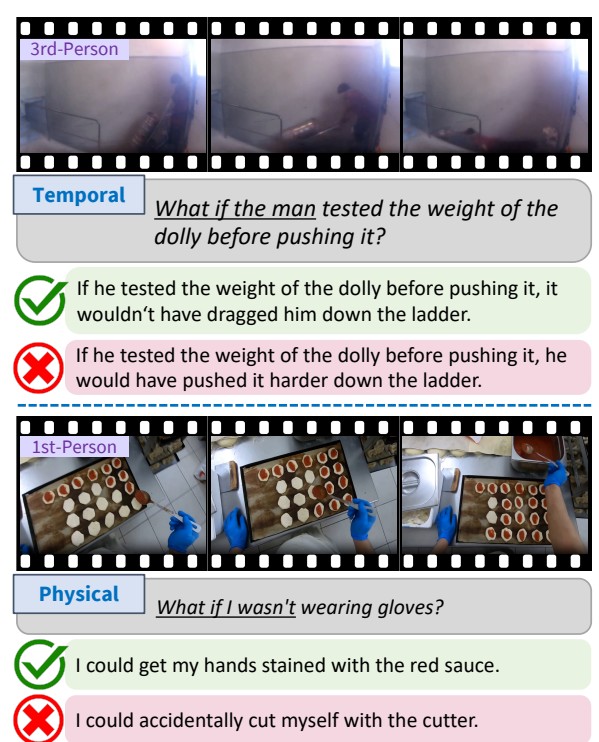

Figure 1: **The ACQUIRED dataset** is a video question answering (QA) dataset that specifically focuses on *counterfactual reasoning* on diverse real-world events. Our dataset concerns three types of commonsense reasoning dimensions: physical, social, and temporal, and encompasses videos from both third-person (upper) and first-person (lower) viewpoints. Each question is curated with a correct and a distractor answer. Each answer is by itself individually judgeable, and hence our dataset can be approached in either binary True/False or multiple-choice setting.

from what actually occurred based on vision and language inputs. It involves mentally simulating alternative scenarios and evaluating their potential outcomes. This cognitive process plays a crucial role in human intelligence, as it allows us to understand causality, make predictions, and learn from past experiences. For AI models, developing the capacity for counterfactual reasoning is a significant area of research and a challenging task. By enabling AI models to engage in counterfactual reasoning, we can enhance their understanding of

---

[*]The authors contribute equally.

causal relationships and their ability to assess the impact of interventions or changes in conditions.

However, despite the significance of counterfactual reasoning, it remains a relatively unexplored area of research. To assess the overall reasoning capabilities of models, several visual question answering datasets have been proposed on both images (Antol et al., 2015; Johnson et al., 2017) and videos (Yi et al., 2020; Xu et al., 2021). These datasets require reasoning skills such as commonsense reasoning, extracting human/object-to-object relations, and inferring physical properties.

One specific dataset in the realm of counterfactual reasoning is CLEVRER (Yi et al., 2020), which generates synthetic videos and associated questions in a controlled environment, featuring simulated object motion and rendered video frames. This dataset allows for evaluating models using descriptive, explanatory, predictive, and counterfactual questions, covering a wide range of reasoning scenarios. However, the data generation process in CLEVRER is overly synthetic, limiting its effectiveness to assess models' counterfactual reasoning abilities in realistic contexts. To address this limitation, TrafficQA (Xu et al., 2021) focuses on real-world traffic event cognition and reasoning in videos, specifically targeting scenarios like traffic accidents. It leverages crowdsourcing to gather diverse types of questions, including fundamental comprehension, counterfactual inference, and event forecasting. Nevertheless, because TrafficQA concentrates solely on traffic events, it fails to encompass other real-life events, resulting in a substantial domain gap between TrafficQA and general video datasets such as Kinetics (Kay et al., 2017; Smaira et al., 2020) and YouTube (Abu-El-Haija et al., 2016; Zellers et al., 2022).

In this paper, we construct a benchmark that can evaluate the counterfactual reasoning abilities of visual models on various kinds of real-world events. We introduce ACQUIRED[1] that covers multiple dimensions of counterfactual reasoning and includes videos of both egocentric and exocentric views. Specifically, based on videos in both Oops (Epstein et al., 2020) and Ego4D (Grauman et al., 2022), we crowd-source 11K questions over 3.7K videos targeting physical, temporal, and social counterfactual reasoning. Both the Oops and Ego4D datasets consist of human activities and interactions in nu-

merous settings, making them ideal sources for curating video question answering datasets. In addition, many videos contain unintentional human actions (*e.g.*, the person accidentally falling down the ladder in Figure 1), which naturally enables people to come up with diverse *what-if* questions.

Inspired by Singh et al. (2021), we adopt a similar methodology for gathering counterfactual questions. Each question consists of a pair of answers, with one being the correct response and the other serving as a distractor. Importantly, the distractor answer represents a *minimal contrastive counterpart* to the correct answer. As we can see from examples in Figure 1, the design of using complementary pairs requires the model to understand the subtle differences between different options, which ensures that the model exhibits an intuitive grasp of counterfactual reasoning. In addition, having one distractor for each question allows for testing models in either True/False or multiple-choice setting.

We extensively evaluate numerous strong language models such as GPT-4, as well as state-of-the-art video-language models such as VALOR on our ACQUIRED dataset. The experimental results suggest that models struggle to effectively utilize the video contexts and perform counterfactual reasoning, with multimodal models achieving only comparable and sometimes inferior performance than language-only models. Moreover, the significant gap between the human and model (>13%) performance highlights the challenging nature of our task and room for improvements in visual counterfactual reasoning.

## 2 Related Work

We will overview three lines of relevant research to this work: visual question answering, visual understanding models, and counterfactual reasoning.

**Visual Question Answering Datasets.** In Table 1, we list several representative visual QA datasets as well as their key features. The Visual Question Answering (VQA) dataset (Antol et al., 2015) is one of the pioneering works in this direction and has been a standard benchmark for evaluating the reasoning ability of image-language models (Goyal et al., 2017). Follow-up datasets such as CLEVR (Johnson et al., 2017) and GQA (Hudson and Manning, 2019) automatically construct compositional questions over real or synthetic images and perform the evaluation in a systematic way. To further evaluate the commonsense reasoning ability of models,

---

[1] Abbreviation of: **A**nswering **C**ounterfactual **Qu**estions **I**n **Re**al-Life **Vid**eos

| Dataset | Visual Source | Question Source | Reasoning Domain | | | Counterfactual |
|---|---|---|---|---|---|---|
| | | | Physical | Temporal | Social | |
| *Image QA datasets* | | | | | | |
| VQA (Antol et al., 2015) | Diverse Real-world Event | Human | ✓ | ✗ | ✗ | ✗ |
| CLEVR (Johnson et al., 2017) | Synthetic Object | Automatic | ✓ | ✗ | ✗ | ✗ |
| GQA (Hudson and Manning, 2019) | Diverse Real-world Event | Automatic | ✓ | ✗ | ✗ | ✗ |
| VCR (Zellers et al., 2019) | Movie | Human | ✓ | ✗ | ✓ | ✗ |
| *Video QA datasets* | | | | | | |
| CLEVRER (Yi et al., 2020) | Synthetic Object Collision | Automatic | ✓ | ✓ | ✗ | ✓ |
| VLEP (Lei et al., 2020) | TV & YouTube | Human | ✓ | ✗ | ✗ | ✗ |
| MovieQA (Tapaswi et al., 2016) | Movie | Human | ✓ | ✓ | ✓ | ✗ |
| MSRVTT-QA (Xu et al., 2017) | Diverse Real-world Event | Automatic | ✓ | ✗ | ✗ | ✗ |
| TGIF-QA (Jang et al., 2017) | Tumblr GIF | Automatic & Human | ✓ | ✓ | ✗ | ✗ |
| MarioQA (Mun et al., 2017) | Gameplay Video | Automatic | ✓ | ✓ | ✗ | ✗ |
| TVQA (Lei et al., 2018) | TV | Human | ✓ | ✓ | ✗ | ✗ |
| Social-IQ (Zadeh et al., 2019) | YouTube | Human | ✗ | ✗ | ✓ | ✗ |
| TrafficQA (Xu et al., 2021) | Traffic Event | Human | ✓ | ✓ | ✗ | ✓ |
| NExT-QA (Xiao et al., 2021) | Diverse Real-world Event | Human | ✓ | ✓ | ✗ | ✗ |
| Causal-VidQA (Li et al., 2022) | Diverse Real-world Event | Human | ✓ | ✗ | ✗ | ✓ |
| ACQUIRED | Diverse Real-world Event | Human | ✓ | ✓ | ✓ | ✓ |

Table 1: **Comparisons** of different visual question answering datasets. ACQUIRED is the first to feature all the dimensions.

VCR (Zellers et al., 2019) crowd-sources common-sense question-answer pairs associated with rationales over static images extracted from movies. Video question answering is more challenging than image question answering and is gaining increasing attention from the research community, leading to several video QA datasets being constructed (Lei et al., 2020; Tapaswi et al., 2016; Xu et al., 2017; Jang et al., 2017; Mun et al., 2017; Lei et al., 2018). Among them, CLEVRER (Yi et al., 2020) improves upon CLEVR and uses programmatically generated videos capturing collisions of synthetic objects to evaluate the model reasoning abilities along multiple dimensions. Social-IQ (Zadeh et al., 2019) and TrafficQA (Xu et al., 2021) employ videos depicting real-world events, wherein Social-IQ primarily emphasizes human social interactions, while TrafficQA focuses on traffic events and accidents. To improve the diversity of the captured events, NExT-QA and Causal-VidQA collect videos from diverse domains and have human-annotated questions targeting different dimensions of reasoning.

As can be seen in Table 1, among all the visual QA datasets, there are only a few that attempt to evaluate the counterfactual reasoning abilities of models. In addition, the existing benchmarks are often limited in terms of the video sources and the question types, making it difficult to evaluate the model performance in a diverse real-world setting. ACQUIRED is the first dataset that can comprehensively evaluate the model counterfactual reasoning abilities spanning three distinct dimensions (*i.e.*, physical, social, and temporal) and cover videos that include a wide range of event types and from different viewpoints.

**Visual Understanding Models.** The creation of visual QA benchmarks allows for the development of visual understanding models. Many of the previous works have tried to solve these tasks using compositional approaches and scene graphs (Santoro et al., 2017; Hu et al., 2017; Hudson and Manning, 2018; Perez et al., 2018; Yi et al., 2018; Shi et al., 2019; Gao et al., 2020; Ding et al., 2021). For example, Hu et al. (2017) propose to train a modular network in an end-to-end manner to achieve both effectiveness and interpretability; Hudson and Manning (2018) utilize scene graphs and perform differentiable neural operations on the graphs to perform visual reasoning. Inspired by the success in pretraining on Internet-scale data (Devlin et al., 2019), pretraining models on large vision and vision-language tasks and then finetuning them on specific downstream tasks has become a standard in tackling visual understanding tasks (Sun et al., 2019; Li et al., 2020; Zhu and Yang, 2020; Lei et al., 2021; Zellers et al., 2021; Fu et al., 2021; Zellers et al., 2022; Wu et al., 2022). Existing works in this direction generally train models on large vision-language datasets with objectives such as masked language modeling and video-text matching. Despite the great progress in this direction, it is unclear if these models can perform counterfactual reasoning. To address this, we benchmark ACQUIRED against state-of-the-art models and systematically study their performance.

**Causal and Counterfactual Reasoning.** Humans can infer how an event would have unfolded differ-

ently without experiencing this alternative reality and it has been a long-standing research topic in cognitive psychology (Van Hoeck et al., 2015). To empower such an important ability to artificial intelligence, researchers have tried to build learning models that can infer causal relations and perform reasoning in various fields (Qin et al., 2019; Yi et al., 2020; Baradel et al., 2020; Abbasnejad et al., 2020; Yue et al., 2021; Wang et al., 2021). Our constructed benchmark provides a valuable resource for developing and evaluating visual models with counterfactual reasoning abilities.

## 3 The ACQUIRED Dataset

### 3.1 Dataset Design & Collection

**Problem Definition.** As illustrated in Figure 1 and Table 2, each data point in ACQUIRED consists of a video and corresponding annotated question and answer pairs. We are inspired by prior works (Clark et al., 2019; Singh et al., 2021) to consider the surprisingly difficult nature of the T/F (yes/no) QA formats that could potentially exhibit less unintended biases/artifacts than curating data in the multiple choice (MCQ) settings. In light of this, for each question, we collect **one correct** and **one distractor** answer (which can be a slightly perturbed version of the correct one), where both of which are individually judge-able by themselves respectively. And hence, our dataset can be approached as a binary *True/False (T/F)* prediction task as well as a *multiple-choice (MCQ)* (2 choices in this case) question answering task.

It is worth noting that the distractors in our dataset are manually curated with certain twists towards the correct answers (examples in Table 2), forcing the models to truly understand the visual concepts involved in the counterfactual questions in order to answer correctly.

In Section 5.2, we will describe our adoption of a pairwise consistency metric that requires the model to answer correctly in both correct and distractor directions to be regarded as a success, in order to reduce the models' exploiting surface-level heuristics to predict the answers.

**Commonsense Dimensions.** We adopt the commonsense knowledge categorization proposed in (Singh et al., 2021), which is inspired by the *Theory of Core Knowledge* (Spelke and Kinzler, 2007)[2], to collect QAs that focus on the following

three dimensions: *physical*, *social*, and *temporal*. The **physical** dimension concerns the knowledge of objects involved in the events and their properties (*e.g.*, shape, size, functionalities, affordances), as well as the motion and location of the events. The **social** dimension looks at human social behaviors, particularly attributes such as personality, emotions, inner interests/intentions, and social activities.[3] The **temporal** dimension regards the aspects of events/activities in their temporal orderings, duration, and frequency/speed of motions.

The three main dimensions are the building blocks towards a comprehensive commonsense reasoning, and helps systematically analyze in which aspects the models need to be improved upon more. Although some questions can be answered using more than one commonsense dimension, we ask the annotators to label with the main one used.

**Video Resources & Sampling.** We utilize the Oops! (Epstein et al., 2020) dataset for third-person view videos and Ego4D (Grauman et al., 2022) for first-person views, where both of which feature text descriptions of the video contents. Oops! concerns predicting the failing (oops) moment of an intended action in a video, and hence is event-rich and a good testbed for reasoning what could the outcomes turned out differently. Ego4D collects videos of humans performing daily activities in the first-person view, which adds a desirable task-knowledge layer on top of its event-richness.

As we are annotating subsets of videos from the aforementioned sources, we have the privilege to encourage a more balanced *key events* distribution from the videos to be annotated. Specifically, we (1) use NLP tools such as semantic role labeling (SRL) to extract key verbs (events) for each video description[4], and group the videos accordingly, (2) each time sample an event group with a probability inverse proportional to the current launched key event distribution, (2) sample a video from the event group in (3), and repeat until reaching a desired number of videos (to be annotated).

The sampling strategy, combined with our predefined reasoning dimensions and video domains, is designed to improve the diversity of question-answer pairs.

**Collection Workflow.** We collect our dataset via

---

[2]The capability of reasoning about physical objects, places, motions, and the social world.

[3]As most videos from Ego4D show tasks performed solely by the camera wearer without social interactions, we do not require the social dimension to be annotated.

[4]We use the originally annotated narrations in Ego4D.

| Sub-sampled Key Video Frames | Question-Answer Pairs |
|---|---|
| | **(Temporal) Q**: What if the two persons had swerved to their left before reaching the shore?
**Correct**: They would not have had a beach landing.
**Wrong**: They would have had a beach landing. |
| | **(Social) Q**: What if the skier was a stranger to the two people standing still?
**Correct**: The skier does not throw the snowball.
**Wrong**: The skier still throws the snowball. |
| | **(Physical) Q**: what if the wheel was in a bike?
**Correct**: He would need to take out the screw before being able to set the wheel on the table
**Wrong**: He would set the whole bike along with the wheel on the table. |
| | **(Physical) Q**: What if I let the cutting board lie on the counter?
**Correct**: The cutting board would be dried slower.
**Wrong**: The cutting board would be dried quicker as it occupies a larger area. |

Table 2: **Sample data points of the ACQUIRED dataset.**

Amazon Mechanical Turk (MTurk). Each MTurk worker is asked to carefully *watch a given video* for creating the QA pairs. As depicted in Figure 2, our dataset collection process comprises four main steps: (1) We design a **qualification questionnaire** focusing on examining one's understanding of the key concepts in our problem design, *i.e.*, the concept of counterfactuality, the requirement of video relevancy, common sense reasoning dimensions, and what types of QA pairs are more desirable. (2) Once the workers pass the qualification test, they are directed to an interface where a **pretrained (text-only) QA model** is deployed in the loop of the QA creation process. Bonus monetary rewards are given if the deployed model fails to predict correctly the creations. (3) Internal members then conduct a **quality validation** on the created samples and provide customized tips and/or feedback to the workers for potential improvements. (4) Lastly, our deployed model is **iteratively finetuned** on the validated samples after each batch of annotations, which results in a constantly improved model to incentivize more challenging sample creations.

Integrating the model-in-the-loop protocol into the pipeline not only brings benefits in curating more challenging samples, but also helps diversify the answers as the models will not be easily fooled if there are similar patterns existed in the dataset or the questions can be simply guessed without visual inputs.

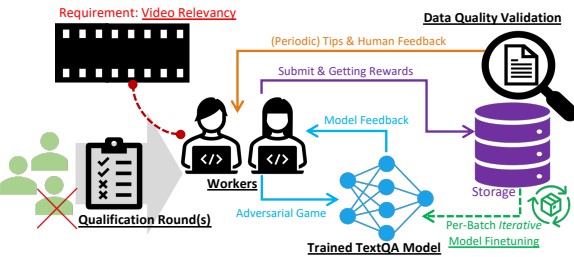

Figure 2: **Data collection workflow.**

**Quality Validation.** In order to further ensure the sample quality as well as summarize common mistakes to provide custom human feedback to the annotators, our internal members conduct the second-phase manual sample validation in conjunction with the deployed model results. We cross-validate the annotations among our internal members in the ramping-up phase to ensure quality. We also accumulate detailed guidelines from our manual validation process for providing effective feedback. After scaling up, we continue to validate the annotations via uniform subsampling across each annotator. Our validation criteria are well aligned as can be seen in the high 0.85 Kappa score for commonsense dimension agreements; and 0.91 overlapping ratios for video relevancy.[5]

---

[5]We did not use Kappa score for video relevancy because there is an unbalanced *"agreed"* distribution of "yes" and "no" (22:1) in our validation results for this criteria, which would result in unfair Kappa score.

| Batches | Annotation Drop-Rate (%) | Number of Videos |
|---|---|---|
| Batch-1 | 28 | 50 |
| Batch-2 | 17.3 | 100 |
| Batch-3 | 4.3 | 200 |
| Batch-4 | 3.5 | 200 |
| Batch-5 | 2.6 | 200 |

Table 3: Annotation drop rate for the first 5 batches. Each video gives 3 pairs of question - correct/distractor answers.

| Type | Counts |
|---|---|
| Total Unique Videos | 2,664 |
| Total Unique QA-Pairs | 7,853 |
| Type-Token Ratio | 0.0158 |
| Verb-Token Ratio (total # verb-types) | 0.0341 |
| Verb-Token Ratio (total # tokens) | 0.0034 |
| Noun-Token Ratio (total # noun-types) | 0.0796 |
| Noun-Token Ratio (total # tokens) | 0.0063 |
| Physical / Social / Temporal (%) | 34 / 33 / 33 |

| Type | Mean | Std | Max | Min |
|---|---|---|---|---|
| Tokens in a Question | 11.5 | 3.5 | 39 | 5 |
| Tokens in an Answer | 8.2 | 6.6 | 60 | 5 |
| Video Frames (Count) | 296.8 | 207.3 | 3283 | 74 |
| Video Duration (sec) | 10.7 | 7.1 | 111.6 | 3.2 |

(a) Videos from Oops!

| Type | Counts |
|---|---|
| Total Unique Videos | 1,038 |
| Total Unique QA-Pairs | 2,695 |
| Type-Token Ratio | 0.0205 |
| Verb-Token Ratio (total # video-types) | 0.0586 |
| Verb-Token Ratio (total # tokens) | 0.0045 |
| Noun-Token Ratio (total # noun-types) | 0.1054 |
| Noun-Token Ratio (total # tokens) | 0.0081 |
| Physical / Social / Temporal (%) | 77 / 0 / 23 |

| Type | Mean | Std | Max | Min |
|---|---|---|---|---|
| Tokens in a Question | 11.1 | 3.3 | 32 | 6 |
| Tokens in an Answer | 9.4 | 6.0 | 41 | 5 |
| Video Frames (Count) | 399.1 | 54.8 | 572 | 240 |
| Video Duration (sec) | 13.3 | 1.8 | 19.3 | 8 |

(b) Videos from Ego4D

Table 4: **General statistics of the two video domains.**

**Validation Analysis.** Table 3 reports the data drop-rates (majority voted to drop by all three validators) for the first 5 batches. We hope these rigorous safety checks can ensure a good data quality that also closely follows our guideline, and the validation should by no means introduce unnecessary biases as we indeed saw a decrease in the dropping rates in our later collection batches.

## 3.2 Dataset Statistics

**General Statistics.** Table 4 summarizes the essential statistics of the collected dataset, where Table 4a is for videos obtained from the Oops! (Epstein et al., 2020) dataset whereas Table 4b is for videos from Ego4D (Grauman et al., 2022). The frame-

| Videos From | Avg. Fool Rate (%) | Avg. Fool Accuracy |
|---|---|---|
| Oops! | 57.69 | 42.31 |
| Ego4D | 51.43 | 48.57 |

Table 5: **Deployed model fooling rates** during collection.

per-second rate (FPS) of videos from either source is mostly 30.

**Key Annotated Events.** We plot the distributions of most frequent key verbs (for main event types) and nouns (for entities involved in events) in Figure 3a and Figure 3b, respectively, to have a rough visual inspection of the diversity of the created samples. The key verbs/nouns are firstly determined by the SRL parses of the question and answer sentences (separately considered), and followed by lemmatization. Both plots are summaries of the two video sources, and more plots broken down by video sources and comparisons with existing works are in Append. Sec. A.2.

**Deployed Model.** Table 5 reports the model fooling rates in our collected data across the two data sources. We encourage our annotators to develop QA pairs that can successfully fool our model by setting up monetary rewards and unlimited trials.

## 4 Benchmarking Models

We benchmark our dataset with both state-of-the-art language-only and vision-language models. Specifically, we perform experiments with DeBERTa (He et al., 2021), UnifiedQA (Khashabi et al., 2020), VIOLET (Fu et al., 2021), VALOR (Chen et al., 2023), and VL-Adapter (Sung et al., 2022) on our dataset.

**Language-Only Models.** While ACQUIRED is a multimodal dataset that has both vision and language inputs, previous works (Thomason et al., 2019) have pointed out that unimodal models can sometimes achieve surprisingly strong performance because of the annotation bias. Therefore, we evaluate both DeBERTa-v3 (He et al., 2021) and the UnifiedQA model family (Khashabi et al., 2020) (state-of-the-art question answering models based on the T5 architecture (Raffel et al., 2020)) on our dataset, which can reflect the dataset biases and provide an important reference point for multimodal models. The language-only models answer the textual questions without looking at the videos.

Inspired by the superior performance of the recent large language models, i.e., the GPT model from OpenAI, we also evaluate its zero-shot perfor-

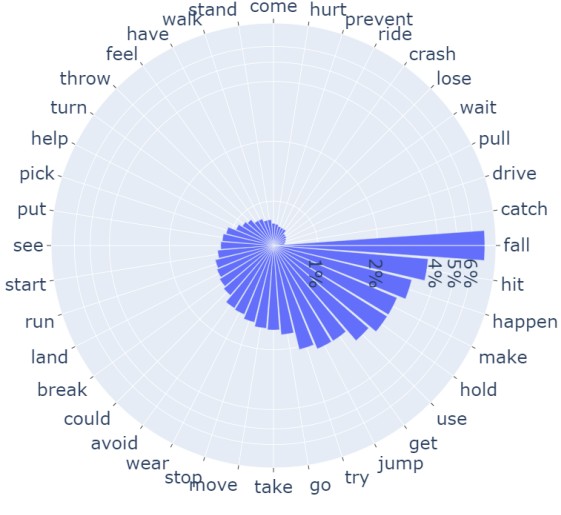

(a) Verbs

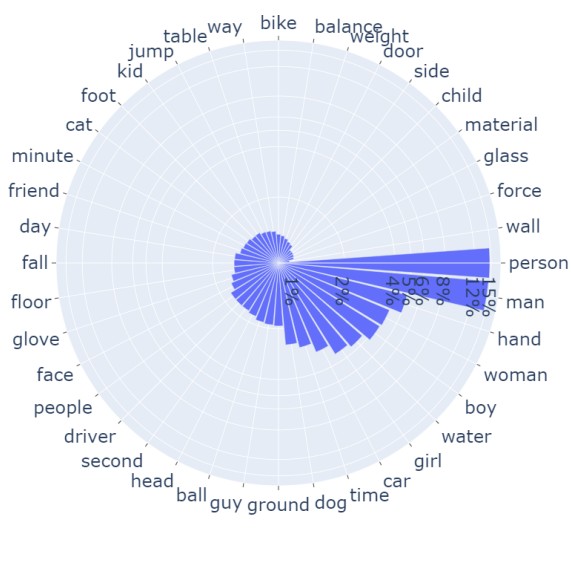

(b) Nouns

Figure 3: **Top-40 frequent word-types** in the dataset.

mance on the textual parts of our dataset. Specifically, we consider both ChatGPT (OpenAI, 2023a) and GPT-4 (OpenAI, 2023b). In addition, we further include a version of GPT models that can condition on pre-annotated descriptions describing the general contents of the videos, to serve as the pseudo visual (and situated) contexts of the questions. Details on how we prompt the GPT family are in Append. Sec. C.1.

**VIOLET (Fu et al., 2021).** VIOLET is a video-language model that has three components, including a video encoder (Swin Transformer-base (Liu et al., 2022)), a language encoder (BERT-base (De-

vlin et al., 2019)), and a cross-modal transformer module that performs cross-modal fusion. The video and language encoders extract features from the video and language inputs respectively, and the extracted features are then fed into the cross-modal transformer for cross-modal interactions. VIOLET is pretrained on large-scale video-text data with masked language modeling that predicts the original word tokens given the masked inputs, masked visual-token modeling (MVM) that recovers the masked video patches conditioned on the unmasked video and language inputs, visual-text matching that aims to align the paired video-text inputs between video and text modality.

**VALOR (Chen et al., 2023).** VALOR is a recently proposed multimodal model that can take video, language, as well as audio as inputs. Similar to VIOLET, VALOR also first encodes vision, audio, and text inputs separately, and the encoded features are then fed into a multimodal decoder for text generation. VALOR demonstrates strong performance across a wide range of tasks, including video retrieval, video captioning, video question answering, audio-visual captioning, text-to-audio retrieval, and audio captioning.

**VL-Adapter (Sung et al., 2022).** VL-Adapter uses a pretrained vision encoder (e.g. CLIP (Radford et al., 2021)) to extract vision features and feed the vision features as well as text tokens to a pretrained language model (e.g. T5 (Raffel et al., 2020)) so that the model can take both vision and language information. When adapting the model for downstream tasks, because it can be costly to finetune all the model parameters, VL-Adapter investigates different adapter-based parameter-efficient finetuning strategies and demonstrates that training the adapter allows them to only update a rather small portion (e.g. 4%) of total parameters and match the performance of finetuning the entire model. Because VL-Adapter supports different combinations of pretrained vision and language encoders, we employ different versions of CLIP-ViT-B/16 and UnifiedQA-Large as the vision and text encoders.

## 5 Experiments and Analysis

### 5.1 Training and Implementation Details

We obtain the pretrained weights of all the benchmarking models from their respective open-sourced releases and finetune them on our official training data split. The hyperparameters are manually tuned

| Modality | Model | QA-Format | Viewpoints | Overall Accuracy↑ (%) | Dimension Breakdowns | | |
| --- | --- | --- | --- | --- | --- | --- | --- |
| | | | | | Physical | Social | Temporal |
| **Text-Only** | DeBERTa-V3 | T/F | — | 70.12 | 70.61 | 70.32 | 69.19 |
| | | MCQ | — | 70.35 | 72.10 | 68.62 | 69.01 |
| | UnifiedQA-base | T/F | — | 68.93 | 70.22 | 69.32 | 66.33 |
| | | MCQ | — | 67.63 | 68.53 | 69.01 | 65.13 |
| | UnifiedQA-large | T/F | — | 69.59 | 71.00 | 69.88 | 67.18 |
| | | MCQ | — | 70.38 | 71.57 | 71.83 | 67.38 |
| | UnifiedQA-3B | T/F | — | 70.49 | 70.58 | 72.20 | 68.99 |
| | | T/F (Pair.) | — | 54.91 | 55.31 | 56.21 | 53.26 |
| | | MCQ | — | 73.40 | 73.36 | 75.80 | 71.60 |
| | Vanilla ChatGPT | T/F | — | 52.80 | 51.36 | 48.06 | 54.04 |
| | Desc.-ChatGPT | T/F | — | 55.20 | 50.82 | 52.90 | 52.48 |
| | | MCQ | — | 42.40 | 36.96 | 43.22 | 47.83 |
| | Vanilla GPT-4 | T/F | — | 53.80 | 53.89 | 53.16 | 54.32 |
| | Desc.-GPT-4 | T/F | — | 56.20 | 55.00 | 58.23 | 55.56 |
| | | MCQ | — | 60.80 | 61.41 | 55.48 | 65.22 |
| **Multimodal** | VIOLET | T/F | All | 66.15 | 70.20 | 64.45 | 60.24 |
| | | T/F (Pair.) | All | 48.25 | 54.03 | 44.60 | 40.63 |
| | | MCQ | All | 69.33 | 70.20 | 70.23 | 67.19 |
| | VALOR | T/F | All | 63.83 | 66.54 | 62.50 | 60.02 |
| | | T/F (Pair.) | All | 43.00 | 46.51 | 42.46 | 37.26 |
| | | MCQ | All | 55.06 | 58.28 | 51.76 | 51.69 |
| | VL-Adapter | T/F | All | 68.75 | 71.56 | 67.94 | 64.40 |
| | | | 3rd | 66.32 | 66.01 | 67.90 | 65.07 |
| | | | 1st | 72.63 | 75.49 | — | 62.82 |
| | | T/F (Pair.) | All | 51.19 | 54.27 | 49.56 | 47.74 |
| | | | 3rd | 47.82 | 47.60 | 49.50 | 46.40 |
| | | | 1st | 60.40 | 62.23 | - | 53.44 |
| | | MCQ | All | 71.53 | 72.70 | 70.39 | 70.25 |
| | | | 3rd | 69.13 | 67.63 | 70.35 | 69.48 |
| | | | 1st | 75.34 | 76.29 | — | 72.05 |
| **Human Performance** | | T/F | All | 83.60 | 81.82 | 100 | 77.27 |
| | | MCQ | All | 92.59 | 90.91 | 100 | 90.91 |

Table 6: **Model benchmarking performance** on our ACQUIRED dataset.

for each model, and the checkpoints used for testing are selected by their validation performance.

## 5.2 Experimental Setup

**Data Splits.** For our official (to-be-released) dataset, we follow a $45-5-50$ ratio and randomly split the train-development-test datasets. The train split is mainly to adapt models to our QA task settings as well as the counterfactual reasoning style. We ensure that there are no overlaps between videos of different sets and the Oops! and Ego4D videos are equally distributed in each of the splits.

**Evaluation Metrics.** Models are evaluated by a simple accuracy metric, for both *T/F* and *MCQ* settings. We also further ablate the model performance along the commonsense dimensions and/or viewpoints, for a more detailed performance breakdown and analysis. We also include the pairwise accuracy in the T/F setting following Singh et al. (2021), where the model is considered correct if both individual judgments are correct in each pair.

**Training Details.** All the models in this work are trained on multi (at least 2-4) Nvidia A100 GPUs[6] on a Ubuntu 20.04.2 operating system.

We train our models until performance convergence is observed on the training split (determined by the development set performance). All of the hyperparameters are manually tuned and searched, with multiple trials for better performance and training convergences.

## 5.3 Experimental Results

Table 6 reports benchmark performance. The best-performing multimodal model (VL-Adapter) performs slightly better than its text-only counterparts, UnifiedQA-large (*i.e.*, the language encoder of our VL-Adapter). While this shows that visual contexts and multimodality are effective, the performance gap is not substantial; therefore, there is

---

[6]https://www.nvidia.com/en-us/data-center/a100/

room for improvement, and more effective methods of multimodal inputs are yet to be explored. While text-only UnifiedQA-3B achieves overall better performance in both T/F and MCQ settings, potentially due to its much larger learnable parameter space, its mediocre pairwise accuracy suggests that the model is still inept at robust counterfactual reasoning in the two facets of the same question.

In general, models perform better in the MCQ settings than the T/F ones. This is intuitive because in the MCQ settings, the model is aware that only one of the two given options is correct and only needs to compare them and select the more reasonable option. (Such a phenomenon is also studied/discovered in (Clark et al., 2019; Singh et al., 2021)) In the case of ChatGPT, its MCQ setting accuracy is lower than that of the T/F setting compared to others. We suspect that ChatGPT might have a weaker reasoning ability compared with GPT4. We observe that often ChatGPT refuses to give an answer in the MCQ settings because of insufficient conditions while it leans towards false when it was asked the same question in a T/F setting.

Perhaps surprisingly, despite the remarkable capabilities of the GPT series, they do not perform as impressively, even when provided with descriptions transcribing the major visual events in the videos. This suggests that the annotators in our curation task indeed closely examine many visual details in order to create more challenging samples.

**Human Performance.** We randomly sub-sample 500 videos to estimate human performance: these are reported in the last two rows of Table 6. The human performance highlights a significant gap above all the model results, especially for the MCQ settings. We hope future modeling endeavors can close the gap in visual counterfactual reasoning.

**Commonsense Dimensions.** The rightmost parts of Table 6 report the performance breakdown along commonsense reasoning dimensions. We observe a general trend: most of the models perform better in physical and social dimensions compared to the temporal dimension; the physical dimension generally exhibits the highest performance. That observation implies that, even after being finetuned on our dataset, the models still fall short of capturing temporal commonsense as opposed to the other two kinds of knowledge. This can also be hypothetically attributed to the fact that the pretraining data for the language models encapsulate more physical

and/or human social knowledge.

**Viewpoints.** We take the best-performing multimodal model (VL-Adapter) and ablate its performance along different video viewpoints. We find that, despite being pretrained mostly on third-person viewpoint videos, the generalization ability of the models towards first-person viewpoints is sufficiently good. However, as the videos from Ego4D are not intended to explicitly contain failed actions from the camera wearers, it could be more challenging for our annotators to construct diverse and subtle counterfactual questions as compared to the videos from Oops!. Nevertheless, we argue that the counterfactual reasoning ability of the models should be equally crucial regardless of video viewpoint, and our dataset can inspire relevant research serving as a first-of-its-kind counterfactual video QA encapsulating videos from varying viewpoints.

## 6 Conclusions

In this work, we present a novel counterfactual-reasoning-focused video question answering dataset, named ACQUIRED. The dataset provides questions about counterfactual hypotheses over visual events (videos). We collect a correct and a distractor answer for three commonsense reasoning dimensions: physical, social, and temporal. We benchmarked various state-of-the-art language models (including LLMs like GPT) and video-language models on the collected dataset, where the results demonstrate algorithm performance well below human performance (>13% accuracy). We hope our studies and the collected ACQUIRED dataset can spur relevant future research, specifically on testing multimodal models' capabilities in counterfactual reasoning, devising assistive AI for remedial and/or cause estimation of observed failures, and more sophisticated visual event understanding and reasoning.

# 7 Limitations

We hereby discuss the potential limitations of our work:

**(1)** Our work focuses on the three commonsense dimensions: physical, social, and temporal. While they likely span the most common types of the reasoning technique, there could be more, *e.g.*, numerical commonsense is not specifically dealt with in this work, nor is non common activities such as fantasies and fictions involved. For future models benchmarked against our dataset, this should be taken account for, *i.e.*, should the models excel at these commonsense dimensions for counterfactual reasoning, we cannot guarantee it is a complete model on all types of reasoning scheme.

**(2)** The videos used in this work are subsets of readily collected ones from both Oops! (Epstein et al., 2020) and Ego4D (Grauman et al., 2022) mother sets, and hence the event distribution can be bounded by the activities they concern. While we argue that the dataset is, to our best knowledge, first of its kind video QA dataset in terms of diversity and dedication of counterfactual reasoning, the video resources spanning even more diversified situations can be further extended. We will release the manuscripts and our collection tools to help spur future relevant research in such endeavours.

**(3)** Unlike Oops!, there is not an obvious failed actions occurred in Ego4D, and hence the annotated questions could be confounded by more imagined situations. We argue that the required reasoning technique is essentially the same and the models learn on our dataset should generalize well to situations that actually involve failing actions from egocentric visual contexts. However, we encourage future research to extend the first-person viewpoint (egocentric) parts to encompass obvious failing actions to collect just-in-time assistive questions and their corresponding remedial responses.

# 8 Ethics and Broader Impacts

We hereby acknowledge that all of the co-authors of this work are aware of the provided *ACL Code of Ethics* and honor the code of conduct. This work is mainly about collecting a sizable video question answering dataset that mainly focuses on counterfactual reasoning abilities and systematically probing such capability from state-of-the-art multimodal and large language models.

**Dataset.** We collect the human annotation of the question-answer pairs to a prompted video via Amazon Mechanical Turk (MTurk) and ensure that all the personal information of the workers involved (e.g., usernames, emails, urls, demographic information, etc.) is discarded in our dataset. Although for a single same video, there is only one single worker annotating the corresponding questions, we ensure that the similar type of events get annotated by as diverse worker pools as possible during the collection phase. We do not foresee much unintended biases within the annotations as they should focus on the contents of the videos for physical and event temporal domains, and we make efforts on reducing the potential biases on the social domain by providing periodic email-based training to the workers to diversify the creations.

This research has been reviewed by the **IRB board** and granted the status of an **IRB exempt**. The detailed annotation process (pay per amount of work, guidelines) is included in the appendix; and overall, we ensure our pay per task is above the the annotator's local minimum wage (approximately $15 USD / Hour). We primarily consider English speaking regions for our annotations as the task requires certain level of English proficiency.

**Techniques.** We benchmark the proposed video counterfactuality reasoning task with the state-of-the-art large-scale pretrained language models (both language-only and multimodal). As the counterfactual commonsense reasoning and its understanding are of our main focus, we do not anticipate production of harmful outputs, especially towards vulnerable populations, after training (and evaluating) models on our proposed task.

# Acknowledgments

Many thanks to J.R. Bronkar for his help on the development of our data curation user interface, and to I-Hung Hsu for his valuable feedback during our paper preparation. This material is based on research supported by the Machine Common Sense (MCS) program under Cooperative Agreement N66001-19-2-4032 and the ECOLE program under Cooperative Agreement HR00112390060, both with the US Defense Advanced Research Projects Agency (DARPA). The views and conclusions contained herein are those of the authors and should not be interpreted as necessarily representing DARPA, or the U.S. Government.

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

## A  Details of The Dataset

Our dataset consists of a mixture of QA pairs collected from two data sources: Ego4d and Oops!. For each dataset split, we create an indexing `.json` file and summarize each QA instance with a video id (index), a domain (physical/social/temporal), a type (counterfactual), a question, a correct answer, a distractor, and a key to the correct answer and a video link URL. Our official data release will encompass all the aforementioned essential fields.

### A.1  Dataset Splits

We split our data into train/val/test based on the ratio 0.45/0.05/0.5, with each unique video only appearing in one split.

### A.2  Word Distributions

Figure 4a and Figure 4b plot the most frequent verbs (mainly for events) and nouns (mainly for entities) distributions of the Oops! proportion of our dataset, while Figure 5a and Figure 5b plot the ones of the Ego4D proportions.

Figure 6a and Figure 6b are distributions of the CLEVRER dataset. Figure 7a and Figure 7b are distributions of the TrafficQA dataset. It can be seen from these charts, alongside Table 7 and Table 8 that the event types in both datasets are quite uni-modally towards their original intended domains (which is reasonable), with all four ratios much lower than those of our dataset.

| Dataset | verb-token ratio | verb-token ratio |
|---------|------------------|------------------|
| CLEVRER | 0.0001 | 5.14e-6 |
| TrafficQA | 0.0053 | 0.0004 |

Table 7: **Verb-token ratio** (total # verb-types / total # tokens) of CLEVRER and TrafficQA

| Dataset | noun-token ratio | noun-token ratio |
|---------|------------------|------------------|
| CLEVRER | 0.0002 | 2.57e-5 |
| TrafficQA | 0.0036 | 0.0006 |

Table 8: **Noun-token** ratio (total # noun-types / total # tokens) of CLEVRER and TrafficQA

## B  Details of Human Annotations

### B.1  Data Validation

We use an internal validation interface with a question-answering setting to accept or reject a sample. This tool also allows us to fix wrong domain categorization and T/F labels annotated by the workers. Specifically, the validation questions include:

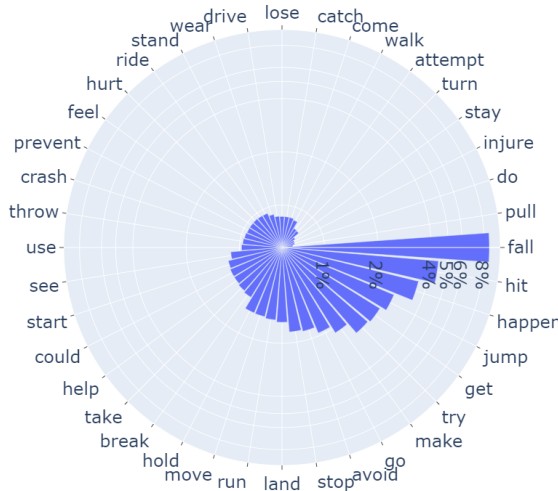

(a) **Top-40 frequent verbs** in Oops! part of ACQUIRED.

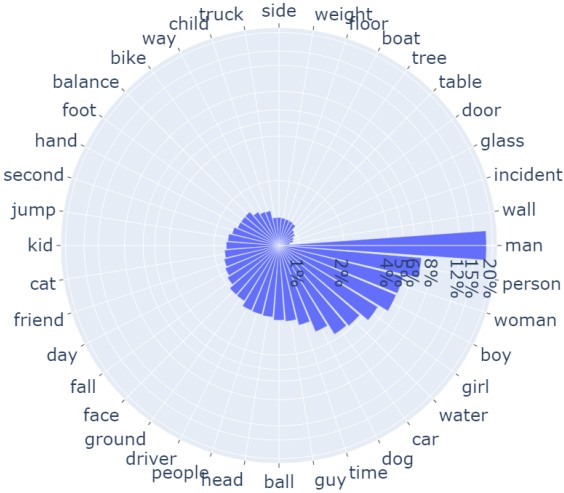

(b) **Top-40 frequent nouns** in Oops! part of ACQUIRED.

Figure 4: **Top-40 frequent word-types** in Oops!.

- Should we discard this question group from our dataset (repetitive / not fixable at all)?

- Does this question group need any editing to reduce ambiguity or to further fool the model?

- Check the T/F of the two sentences.

- Select the domain that you think this question group can be categorized into.

- select one of the type that you think this question group can be categorized into.

- To answer this question, do you need to refer to the video?

- Does this question group conform to our question format?

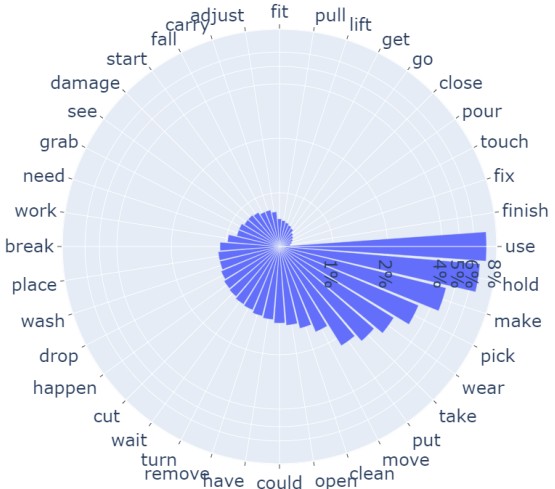

(a) **Top-40 frequent verbs** in Ego4d part of ACQUIRED.

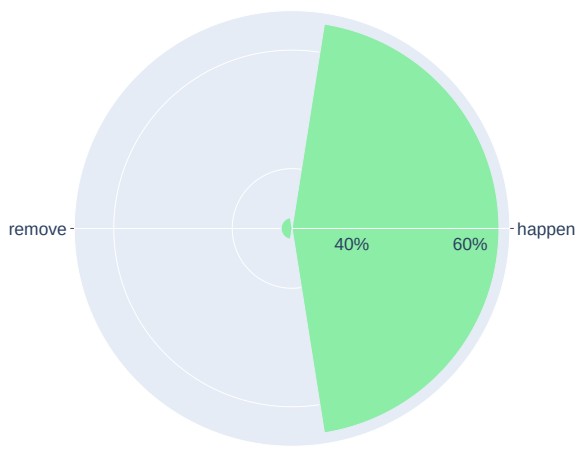

(a) **Top-40 frequent verbs** in CLEVRER.

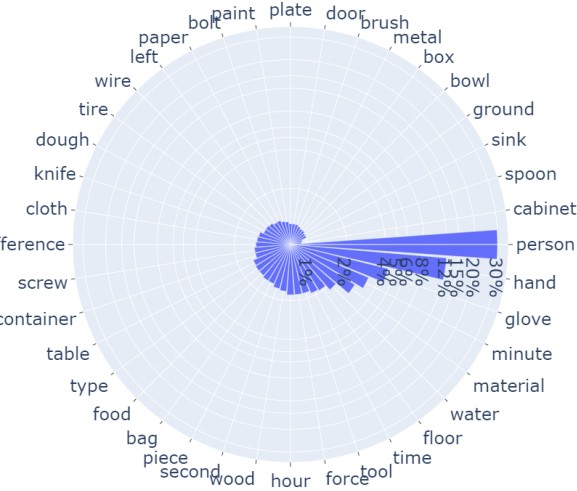

(b) **Top-40 frequent nouns** in Ego4d part of ACQUIRED.

Figure 5: **Top-40 frequent word-types** in Ego4d.

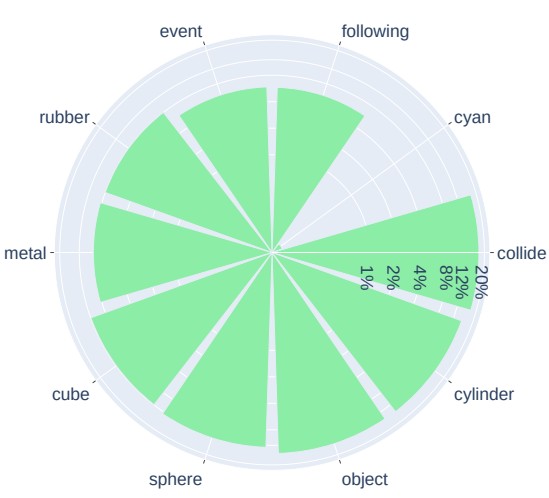

(b) **Top-40 frequent nouns** in CLEVRER.

Figure 6: **Top-40 frequent word-types** in CLEVERER.

## C Modelling Details

### C.1 More on GPT Baselines

The prompts engineered for both ChatGPT and GPT-4 baselines are shown below:

```
# T/F setting without video description
GPT Prompt = The answer to {Question} is
    {Correct/wrong Answer}, True or
    False?

# T/F setting with video description
GPT Prompt = The video is about {Video
    description}, the answer to {
    Question} is {Correct/wrong Answer},
    True or False?

# MCQ setting without video description
GPT Prompt = Which of the following is
    the correct answer to {Question}? (a
    ) answer 1 (b) answer 2
```

## B.2 Annotation Process

We build a user interface to collect QA pair annotations. In addition, we collect human performance as a benchmarking source.

**User Interface.** Our annotation interface (Figure 8b) is launched with Mturk tasks (Figure 8a). Upon accepting each Mturk HIT, our workers will be directed to the annotation web app and do the rest of the task. Workers will be asked to create QA pairs for different domains and assign their T/F labels. If the QA pair successfully fools the model, a green tick will be shown.

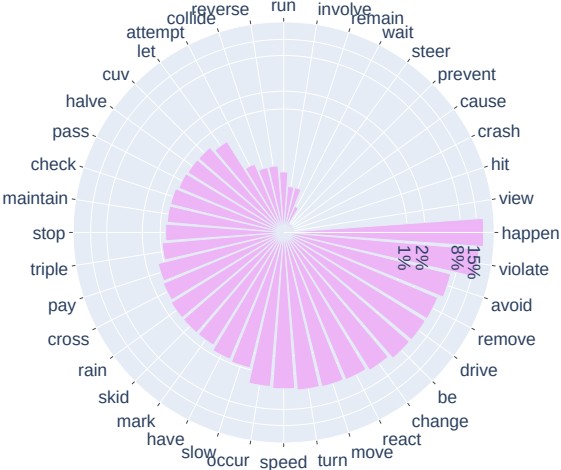

(a) **Top-40 frequent verbs** in TrafficQA.

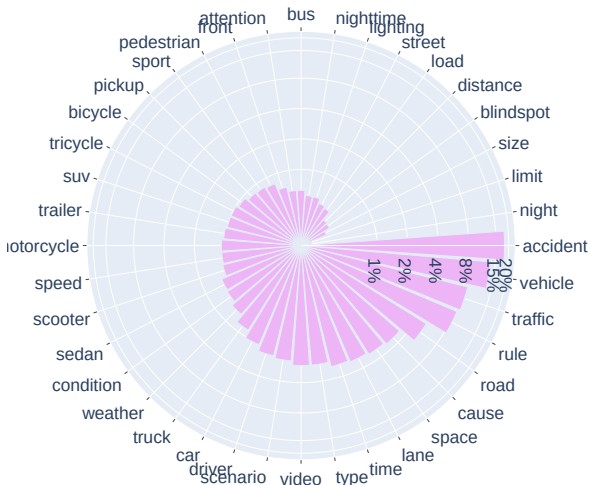

(b) **Top-40 frequent nouns** in TrafficQA.

Figure 7: **Top-40 frequent word-types** in TrafficQA.

```
# MCQ setting with video description
GPT Prompt = The video is about {Video
    description}. Which of the following
    is the correct answer to {Question
    }? (a) answer 1 (b) answer 2
```

## C.2  Training & Implementation Details

**Hyperparameters.** The training time for the text-only models is about 4-5 hours, and it takes about 3-5 hours for multimodal models to converge. We list all the hyperparameters used in Table 9 for the benchmarking models.

**Implementation Details.** The implementations of the transformer-based models are extended from the HuggingFace[7] code base (Wolf et al., 2020),

and our entire code-base is implemented in Py-Torch.[8]

## D  Releases & Code

The comprehensive human-annotated datasets will be released upon acceptance, along with clearly stated documentation for usage. We plan to also release the codes for processing the datasets and leave pointers to all of our benchmarking baseline models. We hope that by sharing the essential resources, our work can incentivize more interest in research on counterfactual reasoning in real-world visually observed events (via the video format), and their applications to robust multimodal AI and assistive technology in both AR and VR.

---

[7]https://github.com/huggingface/transformers

[8]https://pytorch.org/

| Models | Batch Size | Learning Rate | #Training Epochs | #Params |
|---|---|---|---|---|
| DeBERTa-V3 | 8 | 1e-5 | 50 | 86M |
| UnifiedQA-base | 8 | 1e-5 | 50 | 220M |
| UnifiedQA-large | 8 | 1e-5 | 50 | 770M |
| UnifiedQA-3B | 8 | 1e-5 | 50 | 3B |
| VIOLET | 8 | 1e-5 | 20 | 163M |
| VALOR | 16 | 1e-5 | 20 | 479M |
| VL-Adapter | 40 | 3e-4 | 20 | 904M (32M learnable) |

Table 9: **Hyperparameters in this work:** All the models are trained with Adam (Kingma and Ba, 2015). We include the numbers of model parameters in the column of *#Params*.

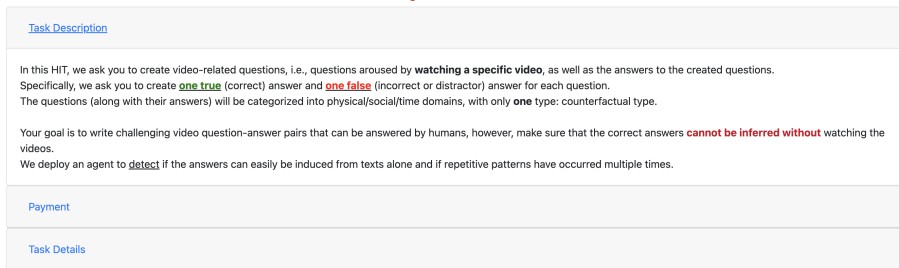

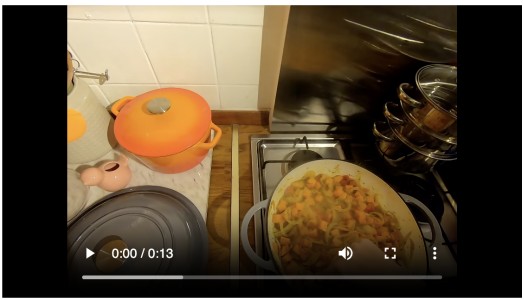

(a) Human Annotation Instruction

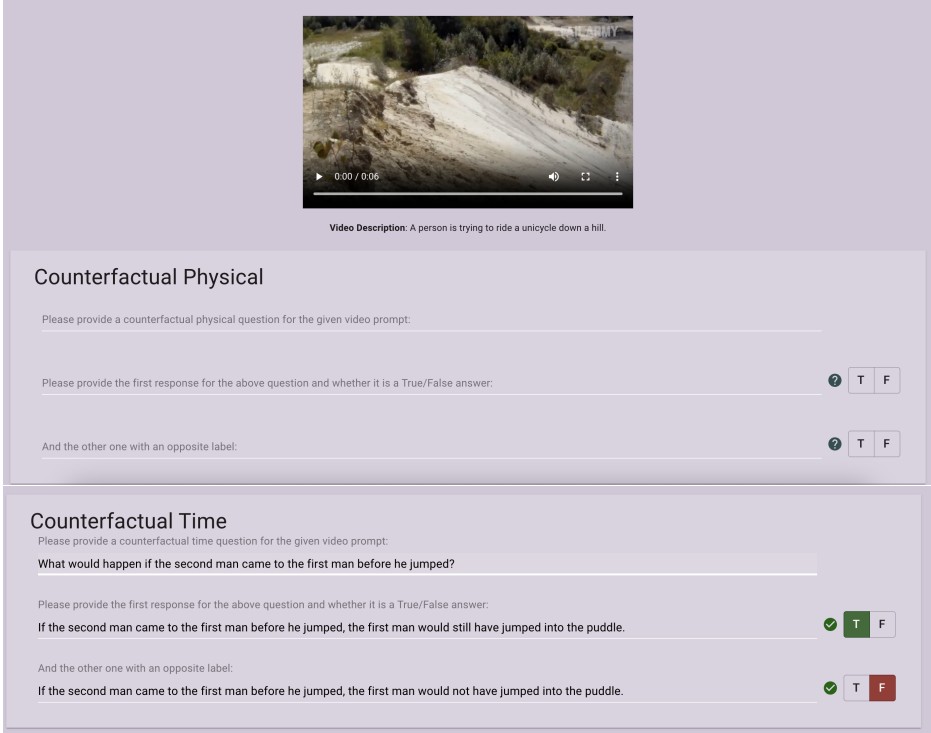

(b) Sample Annotation Interface

Figure 8: **MTurk Annotation User Interface: (a)** We ask workers to follow the indicated instruction. All the blue-colored text bars on the top of the page are expandable. Workers can click to expand them for detailed instructions of the annotation task. **(b)** We design an user-friendly and interactive annotation tool where annotators and simply input their annotations and get an instant feedback from our model.