# OpenReview forum: "ACQUIRED: A Dataset for Answering Counterfactual Questions In Real-Life Videos"
_EMNLP/2023/Conference — EMNLP 2023 Main_

### Official Review · Reviewer_LAU8 · 2023-07-20

**Soundness:** 3

**Excitement:**

4: Strong: This paper deepens the understanding of some phenomenon or lowers the barriers to an existing research direction.

**Missing References:**

- [1] Xiao, Junbin et al. “NExT-QA: Next Phase of Question-Answering to Explaining Temporal Actions.” *2021 IEEE/CVF Conference on Computer Vision and Pattern Recognition (CVPR)* (2021): 9772-9781.
- [2] Grunde-McLaughlin, Madeleine et al. “AGQA: A Benchmark for Compositional Spatio-Temporal Reasoning.” *2021 IEEE/CVF Conference on Computer Vision and Pattern Recognition (CVPR)* (2021): 11282-11292.
- [3] Li, Jiangtong et al. “From Representation to Reasoning: Towards both Evidence and Commonsense Reasoning for Video Question-Answering.” *2022 IEEE/CVF Conference on Computer Vision and Pattern Recognition (CVPR)* (2022): 21241-21250.

**Paper Topic And Main Contributions:**

This paper introduces the ACQUIRED dataset, which focuses on answering counterfactual questions in real-life videos. The dataset incorporates three dimensions of reasoning: physical, social, and temporal. The paper benchmarks several video-language models and SOTA text-only model (GPT4) on the dataset and demonstrates that the models struggle to effectively utilize video contexts and perform counterfactual reasoning.

**Questions For The Authors:**

In section 3.1 Dataset Collection Workflow:
- What pretrained (text-only) QA model do you select and what is your reasoning behind this choice?
- how do you sample for validation? I am interested in details except for the questions,  such as the sample rate and quantitive results like error rate and etc.
In section 3.1 Commonsense Dimensions:
- Why did you categorize the questions as physical, social, and temporal? And the Ego4D dataset contains no social-type questions.

**Reasons To Accept:**

This work present a novel counterfactual reasoning-focused video question answering dataset, named ACQUIRED

**Reasons To Reject:**

- Unclear annotation details, see questions for details.
- You seem to have overlooked some existing VideoQA datasets that involve causality, such as NExTQA [1] and Causal-VidQA [3]. I would appreciate a more in-depth analysis of these.
- A portion of the writing is difficult to understand and requires enhancement

**Reproducibility:**

4: Could mostly reproduce the results, but there may be some variation because of sample variance or minor variations in their interpretation of the protocol or method.

**Reviewer Confidence:**

3: Pretty sure, but there's a chance I missed something. Although I have a good feel for this area in general, I did not carefully check the paper's details, e.g., the math, experimental design, or novelty.

**Typos Grammar Style And Presentation Improvements:**

- Sec. 3.1, line 294: "fails to predict correctly the creations" should be "fails to correctly predict the creations"
- The unnecessarily large margin of related work contrasts with the vague construction procedure of the dataset, as presented in both the papers and the appendix

---

> ### Author Rebuttal · Authors · 2023-08-29
>
> Thank you for the feedback and detailed review!  We are encouraged that you find our dataset novel and exciting for the research community on learning multimodal models that can perform counterfactual reasoning over dynamic visual scenes (videos).
>
> Please find your suggestions and concerns addressed below:
>
> ---
>
> **[W1] Unclear annotation details**
>
> We address all the questions regarding the annotation details below in **[Q]**(s).
>
> ---
>
> **[W2] Comparisons with the existing works: NExT-QA and Causal-VidQA**
>
> We appreciate the reviewer for pointing out additional related works and will discuss in detail below how these works relate to ours, the distinction between them, and our merits compared to the prior literature.
>
> **NExT-QA**, while being a video reasoning dataset, does not focus on counterfactual reasoning, which is our main motivation in this work. The dataset primarily concerns the comprehensive temporal reasoning questions including the cause, the future, and the past of an event of interest shown in a given video. The reasoning ability lies in causality, future prediction, and factual understanding from the video contents.
>
> **Causal-VidQA**, while also having counterfactual video questions included in the dataset similar to previous datasets such as TrafficQA, still differs from our work in multiple aspects. We carefully review the paper and hereby discuss their relations to our work:
>
> - The QAs in ACQUIRED are collected in the three reasoning dimensions (*i.e.*, temporal, physical, social) and two video viewpoints (*i.e.*, both first-person and third-person), beneficial for a more systematic analysis on the models’ abilities in different scenarios and providing the possibility to investigate models in cross-view and cross-domain settings. The aforementioned considerations are uniquely taken into account when collecting our dataset, while Causal-VidQA is based on the Kinetics(-700 2020) dataset where the videos are mostly in the third-person view and its questions are mostly focused on the physical reasoning dimension.
>
> - In addition to the defined dimensions (physical, social, temporal), we further encourage the diversity of question-answer pairs by a designed sampling algorithm (Section 3.1, L267-278). On the other hand, the questions in Causal-VidQA mainly concern the proposed objects by off-the-shelf vision models which are not perfect (*e.g.*, their models can underperform recent works like SAM [4] by a large margin), but also limit the main components the curated questions (and answers/reasons) can be based on. In contrast, we did not impose such a constraint and encourage the annotators to focus on the dynamics/motions of the presented videos and pay attention to the subtle visual details when constructing the counterfactuality. The comparison can be elaborated by the following metrics:
>   - **Type-Token Ratio** is the ratio obtained by dividing the types (unique vocabulary) by the total number of words (higher ratio indicates higher degree of lexical variations). We compute the ratios for both verbs and nouns, against both total verbs/nouns and total words.
> | **Datasets** | **Verb-Token Ratio (Total Tokens)** | **Verb-Token Ratio (Total Verbs)** | **Noun-Token Ratio (Total Tokens)** | **Noun-Token Ratio (Total Nouns)** |
> | :--------: | :--------: | :--------: | :--------: | :--------: |
> | Causal-VidQA | 0.000907 | 0.00727 | 0.001750 | 0.02192 |
> | Causal-VidQA (w/o "happen") | 0.0009198 | 0.008164 | --- | --- |
> | ACQUIRED | 0.004651 | 0.06874 | 0.0089 | 0.1133 |
>   - Note that we observed that many of the Causal-VidQA questions have the word (verb) “happen” in them, and hence we additionally compute the ratios excluding those *verbs*.
>   - It can be seen that the ACQUIRED dataset indeed contains more diverse action/interaction scenarios, *i.e.*, verb ratios; and the relationship to the distinct objects, *i.e.*, noun ratios, both exhibit much higher ratio scores.
>
> - We have an iterative model-in-the-loop data collection protocol that can mitigate the annotator biases introduced in the annotation process (L279-302). The advantage of this protocol can be demonstrated by the text-only model performance as reported below (we report the Q → A results on the Causal-VidQA validation set as the test set GTs are not provided).  We can see rather significant performance improvements of text-only models compared with the random baseline on Causal-VidQA while much less improvements on ours, demonstrating that the iterative model-in-the-loop data collection protocol not only brings benefits in curating more challenging samples, but also helps diversify the answers as the models will not be easily fooled if **repetitiveness is too severe** (among the dataset) or the questions can be **simply guessed with commonsense** (no video requirement).
>   - **GPT performance is *zero-shot***
>   - The descriptions (desc.) of Causal-VidQA are directly obtained from the original Kinetics-700 dataset, which are essentially short phrases of the video activity classes (labeled for activity recognition).
> | **Datasets** | **Model** | **QA-Format** | **Accuracy (%)** | **Gain (from Random) (%)** |
> | :--------: | :--------: | :--------: | :--------: | :--------: |
> | Causal-VidQA | Random | MCQ | 20.0 | 0 |
> | Causal-VidQA | Vanilla ChatGPT | MCQ | 42.0 | 110 |
> | Causal-VidQA | Vanilla GPT-4 | MCQ | 48.8 | 144 |
> | Causal-VidQA | Desc.-GPT-4 | MCQ | 52.6 | 163 |
> | Causal-VidQA | DeBERTa-v3 | MCQ | 66.2 | 231 |
> | Causal-VidQA | UnifiedQA-3B | MCQ | 60.7 | 204 |
> | ACQUIRED | Random | T/F | 50.0 | 0 |
> | ACQUIRED | Vanilla ChatGPT | T/F | 52.8 | 5.6 |
> | ACQUIRED | Vanilla GPT-4 | T/F | 53.8 | 7.6 |
> | ACQUIRED | Desc.-GPT-4 | T/F | 56.2 | 12.4 |
> | ACQUIRED | DeBERTa-v3 | T/F  | 70.12 |  40.2 |
> | ACQUIRED | UnifiedQA-3B  | T/F | 70.49 | 41.9 |
>
> - We collect more challenging distractors than Causal-VidQA. The distractors in our dataset are manually curated with certain twists towards the correct answers (examples in Table 2), forcing the models to truly understand the visual concepts involved in the counterfactual questions. Furthermore, we also report the pairwise accuracy to evaluate the **true robustness** of the models. On the other hand, the distractors in Causal-VidQA are retrieved from answers in similar questions.
>
> - In summary, while sharing high-level similarities, our work expands the views of counterfactual reasoning into more comprehensive (in terms of knowledge dimensions) and diverse (in terms of QA semantics and events) regimes. In addition, we spend additional efforts on ensuring the diversity and difficulty of our constructed dataset. In light of this, we do not think the existence of such prior work should exclude/diminish the importance of our work, towards the research community. That said, we thank the reviewer for pointing out the two related works and we will be sure to add them to corresponding sections with the discussion above.
>
> [4] Kirillov, Alexander, et al. "Segment anything." *ICCV* 2023
>
> ---
>
> **[W3] Presentation improvements**
>
> Thank you, we will improve on our paper presentation in the final version.
> We would also appreciate that the reviewer could more specifically point out or suggest which parts need more revisions and we will be sure to revise them accordingly.
>
> ---
>
> **[Q1] Pretrained text-only QA model**
>
> As mentioned in L358-365 in Section 4, beside DeBERTa-v3, the strongest text-only model we adopt is the UnifiedQA model [5ab] (specifically its T5-3B variant), mainly for both its computational efficiency and proven strong performance in many QA tasks (please refer to their original paper, where the model is extensively tested on many existing QA benchmarks).
>
> [5a] Khashabi, Daniel, et al. "Unifiedqa: Crossing format boundaries with a single qa system." *Findings of EMNLP* 2020
>
> [5b] Khashabi, Daniel, Yeganeh Kordi, and Hannaneh Hajishirzi. "Unifiedqa-v2: Stronger generalization via broader cross-format training." *arXiv preprint* 2022
>
> ---
>
> **[Q2] Validation phase details**
>
> For the first few annotation batches, we sample all of the curated data for manual validation for both the data quality control as well as consolidating feedback to further train the annotators for our task.
> From the 4th batch where we deem the annotators well-trained, we sample 20% of the data for efficiency and the error rate is no higher than 3-4%, judging by our desired data criteria (*i.e.*, whether the created question is counterfactual, whether the common sense dimension is reasonable, and whether the question indeed requires the video, etc.).
>
> Below we report the data drop-rates (*majority voted to drop* by all three validators) for the first 5 batches (note that each video gives 3 pairs of question- correct/distractor answers):
> | **Batches** | **Batch-1** | **Batch-2** | **Batch-3** | **Batch-4** | **Batch-5** |
> | :--------: | :--------: | :--------: | :--------: | :--------: | :--------: |
> | Annotation Drop-Rate (%) | 28 | 17.3 | 4.3 | 3.5 | 2.6 |
> | Number of Videos | 50 | 100 | 200 | 200 | 200 |
>
> ---
>
> **[Q3] Consideration on the common sense dimensions**
>
> We are inspired by the commonsense benchmark prior work [6], where they are inspired by the **Theory of Core Knowledge** [7], *i.e.*, the capability of reasoning about physical objects, places, motions, and the social world. The three main dimensions are the building blocks towards a comprehensive commonsense reasoning, and helps systematically analyze in which aspects the models need to be improved upon more.
>
> Our primary motivation is to provide a dataset where a model can learn to reason over counterfactuality from different viewing angles, as the learned agent can be robust in assisting humans in both 1st and 3rd person viewpoints. Our initial preliminary studies found that Ego4D, as an egocentric human activity recorded video set, contains much more generic task completions than social interactions (many of the videos only have activities performed by a single person), and hence our annotators found it more difficult than Oops! to annotate the social dimension. We hence made the decision to mainly collect Physical and Time which are more prevalent in the Ego4D set (in this work).
>
> However, we do not expect such design choices would hinder the models from learning from the three main commonsense dimensions, as there is still a decent amount of social portion from the Oops! Videos.
>
> [6] Singh, Shikhar, et al. "COM2SENSE: A commonsense reasoning benchmark with complementary sentences." *Findings of ACL* 2021
>
> [7] Spelke, Elizabeth S., and Katherine D. Kinzler. "Core knowledge." *Developmental science* 10.1 (2007): 89-96.
>
> ---
>
> **[Misc.] Missing references and typos**
>
> We will cite the suggested references appropriately in our final version. Specifically for paper [1] and [3], we will include the discussion mentioned above. Thanks again for the presentation suggestion, we will fix the typos and enrich the related works section aiming at a better clarity.

---

### Official Review · Reviewer_5Kzu · 2023-08-02

**Soundness:** 3

**Excitement:**

3: Ambivalent: It has merits (e.g., it reports state-of-the-art results, the idea is nice), but there are key weaknesses (e.g., it describes incremental work), and it can significantly benefit from another round of revision. However, I won't object to accepting it if my co-reviewers champion it.

**Paper Topic And Main Contributions:**

This paper contribute a VideoQA dataset for Counterfactual Questions.
To construct this dataset, the author propose to provide a first- and a third-person viewpoint of the video, and provide the answer with a wrong one and a correct one.
Experimental results shows that there are still a long way to go in this field.

**Questions For The Authors:**

Refer to Reasons To Reject

**Reasons To Accept:**

1. This work propose a VideoQA dataset for Counterfactual Questions.
2. This work propose extensive experimental results to show how existing model perform on this dataset.

**Reasons To Reject:**

1. The novelty of this work is quite limited. Since similar dataset, like Causal-VidQA[1] and NeXT-QA[2] have already existed.
2. Missing discussion with similar datasets [1][2]
3. The answer only includes a wrong and and a right answer, however, choosing the right answer don't indicate that the model can really reasoning in counterfactual situation. A more suitable solution is to make the answer process as answer-reason mode. Please refer to [1] and VCR for more details.

[1] From representation to reasoning: Towards both evidence and commonsense reasoning for video question-answering, CVPR2022
[2] Next-qa: Next phase of question-answering to explaining temporal actions. In CVPR 2022

----
After rebuttal:
The author partial solves my concerns, so I will also slightly increase my rating.

**Reproducibility:**

5: Could easily reproduce the results.

**Reviewer Confidence:**

5: Positive that my evaluation is correct. I read the paper very carefully and I am very familiar with related work.

---

> ### Author Rebuttal · Authors · 2023-08-29
>
> We thank the reviewer for the feedback and the constructive suggestions! We appreciate that the reviewer agrees that our dataset is useful for learning multimodal models on the counterfactual reasoning ability, and our extensive benchmarking studies helpful for understanding where the proposed dataset stands.
>
> Please find your suggestions and concerns addressed below:
>
> ---
>
> **[W1,2] Comparisons with the existing works: NExT-QA[2] and Causal-VidQA [1]**
>
> Thanks for pointing out the related works!
> We will cite these works and compare ours with them in detail in the revised version.
> Below we discuss how these works are different from ours, and our merits compared to the prior literature.
>
> **NExT-QA**, while being a video reasoning dataset, does not focus on counterfactual reasoning, which is our main motivation in this work. The dataset primarily concerns the comprehensive temporal reasoning questions including the cause, the future, and the past of an event of interest shown in a given video. The reasoning ability lies in causality, future prediction, and factual understanding from the video contents.
>
> **Causal-VidQA**, while also having counterfactual video questions included in the dataset similar to previous datasets such as TrafficQA, still differs from our work in multiple aspects. We carefully review the paper and hereby discuss their relations to our work:
>
> - The QAs in ACQUIRED are collected in the three reasoning dimensions (*i.e.*, temporal, physical, social) and two video viewpoints (*i.e.*, both first-person and third-person), beneficial for a more systematic analysis on the models’ abilities in different scenarios and providing the possibility to investigate models in cross-view and cross-domain settings. The aforementioned considerations are uniquely taken into account when collecting our dataset, while Causal-VidQA is based on the Kinetics(-700 2020) dataset where the videos are mostly in the third-person view and its questions are mostly focused on the physical reasoning dimension.
>
> - In addition to the defined dimensions (physical, social, temporal), we further encourage the diversity of question-answer pairs by a designed sampling algorithm (Section 3.1, L267-278). On the other hand, the questions in Causal-VidQA mainly concern the proposed objects by off-the-shelf vision models which are not perfect (*e.g.*, their models can underperform recent works like SAM [3] by a large margin), but also limit the main components the curated questions (and answers/reasons) can be based on. In contrast, we did not impose such a constraint and encourage the annotators to focus on the dynamics/motions of the presented videos and pay attention to the subtle visual details when constructing the counterfactuality. The comparison can be elaborated by the following metrics:
>   - **Type-Token Ratio** is the ratio obtained by dividing the types (unique vocabulary) by the total number of words (higher ratio indicates higher degree of lexical variations). We compute the ratios for both verbs and nouns, against both total verbs/nouns and total words.
> | **Datasets** | **Verb-Token Ratio (Total Tokens)** | **Verb-Token Ratio (Total Verbs)** | **Noun-Token Ratio (Total Tokens)** | **Noun-Token Ratio (Total Nouns)** |
> | :--------: | :--------: | :--------: | :--------: | :--------: |
> | Causal-VidQA | 0.000907 | 0.00727 | 0.001750 | 0.02192 |
> | Causal-VidQA (w/o "happen") | 0.0009198 | 0.008164 | --- | --- |
> | ACQUIRED | 0.004651 | 0.06874 | 0.0089 | 0.1133 |
>   - Note that we observed that many of the Causal-VidQA questions have the word (verb) “happen” in them, and hence we additionally compute the ratios excluding those *verbs*.
>   - It can be seen that the ACQUIRED dataset indeed contains more diverse action/interaction scenarios, *i.e.*, verb ratios; and the relationship to the distinct objects, *i.e.*, noun ratios, both exhibit much higher ratio scores.
>
> - We have an iterative model-in-the-loop data collection protocol that can mitigate the annotator biases introduced in the annotation process (L279-302). The advantage of this protocol can be demonstrated by the text-only model performance as reported below (we report the Q → A results on the Causal-VidQA validation set as the test set GTs are not provided).  We can see rather significant performance improvements of text-only models compared with the random baseline on Causal-VidQA while much less improvements on ours, demonstrating that the iterative model-in-the-loop data collection protocol not only brings benefits in curating more challenging samples, but also helps diversify the answers as the models will not be easily fooled if **repetitiveness is too severe** (among the dataset) or the questions can be **simply guessed with commonsense** (no video requirement).
>   - **GPT performance is *zero-shot***
>   - The descriptions (desc.) of Causal-VidQA are directly obtained from the original Kinetics-700 dataset, which are essentially short phrases of the video activity classes (labeled for activity recognition).
> | **Datasets** | **Model** | **QA-Format** | **Accuracy (%)** | **Gain (from Random) (%)** |
> | :--------: | :--------: | :--------: | :--------: | :--------: |
> | Causal-VidQA | Random | MCQ | 20.0 | 0 |
> | Causal-VidQA | Vanilla ChatGPT | MCQ | 42.0 | 110 |
> | Causal-VidQA | Vanilla GPT-4 | MCQ | 48.8 | 144 |
> | Causal-VidQA | Desc.-GPT-4 | MCQ | 52.6 | 163 |
> | Causal-VidQA | DeBERTa-v3 | MCQ | 66.2 | 231 |
> | Causal-VidQA | UnifiedQA-3B | MCQ | 65.9 | 230 |
> | ACQUIRED | Random | T/F | 50.0 | 0 |
> | ACQUIRED | Vanilla ChatGPT | T/F | 52.8 | 5.6 |
> | ACQUIRED | Vanilla GPT-4 | T/F | 53.8 | 7.6 |
> | ACQUIRED | Desc.-GPT-4 | T/F | 56.2 | 12.4 |
> | ACQUIRED | DeBERTa-v3 | T/F  | 70.12 |  40.2 |
> | ACQUIRED | UnifiedQA-3B  | T/F | 70.49 | 41.9 |
>
> - We collect more challenging distractors than Causal-VidQA. The distractors in our dataset are manually curated with certain twists towards the correct answers (examples in Table 2), forcing the models to truly understand the visual concepts involved in the counterfactual questions. Furthermore, we also report the pairwise accuracy to evaluate the **true robustness** of the models. On the other hand, the distractors in Causal-VidQA are retrieved from answers in similar questions.
>
> - In summary, while sharing high-level similarities, our work expands the views of counterfactual reasoning into more comprehensive (in terms of knowledge dimensions) and diverse (in terms of QA semantics and events) regimes. In addition, we spend additional efforts on ensuring the diversity and difficulty of our constructed dataset. In light of this, we do not think the existence of such prior work should exclude/diminish the importance of our work, towards the research community. That said, we thank the reviewer for pointing out the two related works and we will be sure to add them to corresponding sections with the discussion above.
>
> [3] Kirillov, Alexander, et al. "Segment anything." *ICCV* 2023
>
> ---
>
> **[W3] The QA format and the complexity of our ACQUIRED dataset**
>
> We would like to point out that our evaluation protocol is beyond the traditional T/F evaluation paradigm and is rather challenging for neural models that are vulnerable to exploit spurious input-output correlations.
>
> We are inspired by prior works [4,5] to consider the **surprisingly difficult nature of the T/F (yes/no) QA formats** that could potentially exhibit less unintended biases/artifacts than curating data in the multiple choice (MCQ) settings.
> As shown in Table 2, we focus on curating high-quality correct answers and a strong associated distractor answer that should be a **twist on the correct** one focusing on similar concepts (of the asked questions).
>
> In addition to having the manually curated distractors, we provide two formulations and two main metrics to evaluate if the models indeed perform correct reasoning instead of utilizing spurious correlations.
> As mentioned in L230-237 in Section 3.1, the T/F format requires the model to provide a yes/no response to **each of the** correct answer and the distractor, where both **standard accuracy metric** and the **pairwise consistency metric** (see L448-451 in Section 5.2) are used.
>
> The pairwise consistency metric thus requires the model to answer correctly **in both correct and distractor directions** to be regarded as a success. Therefore, simply using surface-level heuristics will fail as the models will output the same prediction given two minimally different statements.
>
> Furthermore, as mentioned above, the distractors in our dataset are also *curated* that undergo the same model-in-the-loop gamified setup, where the aforementioned **pairwise accuracy** can more robustly evaluate whether the models **fully understand** the contexts and are able to reason over them.
> Judging by the much deteriorated numbers (from Table 5) of the pairwise metric as compared to the standard ones, there indeed is much room for improvements to make the models more robust on their beliefs (they should be close to each other if the models **truly** understand the counterfactual contexts).
>
> While we acknowledge the importance and the interestingness of asking the models to provide the rationale behind their decisions, it is still an open question whether this setting can reveal if models can indeed perform reasoning because this is yet another MCQ setting, while our **pairwise accuracy** goes beyond this paradigm and is designed to evaluate if models can robustly perform correct reasonings.
>
> [4] Clark, Christopher, et al. "BoolQ: Exploring the surprising difficulty of natural yes/no questions." *NAACL-HLT* 2019
>
> [5] Singh, Shikhar, et al. "COM2SENSE: A commonsense reasoning benchmark with complementary sentences." *Findings of ACL* 2021

---

### Official Review · Reviewer_sY4g · 2023-08-02

**Soundness:** 4

**Excitement:**

4: Strong: This paper deepens the understanding of some phenomenon or lowers the barriers to an existing research direction.

**Paper Topic And Main Contributions:**

The paper proposes a new real-life dataset, ACQUIRED, for multimodal counterfactual reasoning. The proposed dataset enables evaluating multi-modal models' ability to do multimodal counterfactual reasoning. The dataset is a collection of video segments annotated with a counterfactual question, and +ve and -ve answer. The authors propose a video question answering question task on this dataset, and benchmark recent models on the task. There is substantial gap between the human and model performance suggesting that the proposed task on the dataset is a challenging one.

**Reasons To Accept:**

1. This is an interesting dataset which can be used as a testbed evaluate models ability to do counterfactual reasoning.
2. The task is challenging with significant performance gap between recent models and human.
3. The dataset would motivate research in the direction of building multimodal models with better counterfactual reasoning capability.
4. Benchmarking is quite extensive and covers very recent models.

**Reasons To Reject:**

1. The internal manual validation/correction phase may alter the data distribution but is necessary to filter out noisy responses.
The author's quality control process is thorough, and the high inter-validator agreement score shows that the validators in general have high agreement on the validation tasks.

**Reproducibility:**

4: Could mostly reproduce the results, but there may be some variation because of sample variance or minor variations in their interpretation of the protocol or method.

**Reviewer Confidence:**

3: Pretty sure, but there's a chance I missed something. Although I have a good feel for this area in general, I did not carefully check the paper's details, e.g., the math, experimental design, or novelty.

---

> ### Author Rebuttal · Authors · 2023-08-29
>
> We thank the reviewer for the insightful remarks! We are encouraged that you find our dataset interesting and useful for relevant research directions. We are particularly delighted to know that you find the resource would benefit building future multimodal models on counterfactual reasoning, and that our initial benchmarking results serve as a good guideline for the future research.
>
> Please find your suggestions and concerns addressed below:
>
> ---
>
> **[W1] Internal validation causing altered data distribution and artifacts/biases**
>
> Our internal validation is meant for assuring the quality of the curated data, and mainly conducted to **filter out significant violations to our desired data criteria while introducing minimal potentially biased expert knowledge into the constructed video-question-answer data points**.
> The primary criteria are: **(1)** whether the QA is truly counterfactual, **(2)** whether the video source is indeed required to solve the QA, and **(3)** are the labeled answers correct and do the classified dimensions make sense (see Section B.1 for more details)?
> Our criteria are *well aligned with some of our preliminary modeling studies*, where the data points deemed not requiring the videos indeed can be more easily exploited by language-only models.
>
> Moreover, we prohibit too subjective judgements by listing the criteria prior to the validation phase, and all the validators are well-trained to fully understand the main criteria/motivations of our dataset collection process, before they can perform the validation tasks. We have three validators which also exhibit high agreements on their judgements towards the quality of the data, please refer to L315-318 and Section B.1 in the appendix.
>
> Furthermore, to ensure the diversity and mitigate the biases, we deployed a model-in-the-loop approach to incentivize more challenging and diverse QA creations, with an event-balancing sampling algorithm (see L267-278 in Section 3.1) to provide the video data points, and hence the data distribution should reflect these implementations more than our subsampled validation process.
>
> Below we report the data drop-rates (*majority voted to drop* by all three validators) for the first 5 batches (note that each video gives 3 pairs of question- correct/distractor answers):
> | **Batches** | **Batch-1** | **Batch-2** | **Batch-3** | **Batch-4** | **Batch-5** |
> | :--------: | :--------: | :--------: | :--------: | :--------: | :--------: |
> | Annotation Drop-Rate (%) | 28 | 17.3 | 4.3 | 3.5 | 2.6 |
> | Number of Videos | 50 | 100 | 200 | 200 | 200 |
>
> We hope these rigorous safety checks can ensure a good data quality that also closely follows our guideline, and the validation should by no means introduce unnecessary biases as we indeed saw a decrease in the dropping rates in our later collection batches.

---

### Meta-Review · Area_Chair_JudR · 2023-09-15

**Recommendation:** 5

**Metareview:**

Three reviewers provided feedback for this work and were in consensus. They found the newly proposed dataset to be interesting and very useful to the community. The task is challenging and need for such datasets is quite high. The reviewers expect that this would be an impactful dataset and would motivate work in the direction of multimodal models with counterfactual reasoning capabilities. They also found the benchmarking to be thorough and well done. The one major concern was positioning the dataset in light of recent works at CVPR 2022. The authors have responded well and have agreed to add these extensive details into the paper. In light of these reviews and discussion, I recommend acceptance.

---

### Decision · Program_Chairs · 2023-10-07

**Decision:**

Accept-Main

**Comment:**

Three reviewers provided feedback for this work and were in consensus. They found the newly proposed dataset to be interesting and very useful to the community. The task is challenging and need for such datasets is quite high. The reviewers expect that this would be an impactful dataset and would motivate work in the direction of multimodal models with counterfactual reasoning capabilities. They also found the benchmarking to be thorough and well done. The one major concern was positioning the dataset in light of recent works at CVPR 2022. The authors have responded well and have agreed to add these extensive details into the paper. In light of these reviews and discussion, I recommend acceptance.